# Altered Emotional Phenotypes in Chronic Kidney Disease Following 5/6 Nephrectomy

**DOI:** 10.3390/brainsci11070882

**Published:** 2021-06-30

**Authors:** Yeon Hee Yu, Seong-Wook Kim, Dae-Kyoon Park, Ho-Yeon Song, Duk-Soo Kim, Hyo-Wook Gil

**Affiliations:** 1Department of Anatomy, College of Medicine, Soonchunhyang University, Cheonan-si 31151, Korea; yuyeon0220@naver.com (Y.H.Y.); mdeornfl@sch.ac.kr (D.-K.P.); 2Graduate School of New Drug Discovery & Development, Chungnam National University, Daejeon 34134, Korea; seongwook0205@gmail.com; 3Department of Microbiology and Immunology, College of Medicine, Soonchunhyang University, Cheonan-si 31151, Korea; songmic@sch.ac.kr; 4Department of Internal Medicine, Soonchunhyang University Cheonan Hospital, Cheonan-si 31151, Korea

**Keywords:** chronic kidney disease, emotional phenotypes, hippocampus, astrogliosis, uremia, fibrosis

## Abstract

Increased prevalence of chronic kidney disease (CKD) and neurological disorders including cerebrovascular disease, cognitive impairment, peripheral neuropathy, and dysfunction of central nervous system have been reported during the natural history of CKD. Psychological distress and depression are serious concerns in patients with CKD. However, the relevance of CKD due to decline in renal function and the pathophysiology of emotional deterioration is not clear. Male Sprague Dawley rats were divided into three groups: sham control, 5/6 nephrectomy at 4 weeks, and 5/6 nephrectomy at 10 weeks. Behavior tests, local field potentials, and histology and laboratory tests were conducted and investigated. We provided direct evidence showing that CKD rat models exhibited anxiogenic behaviors and depression-like phenotypes, along with altered hippocampal neural oscillations at 1–12 Hz. We generated CKD rat models by performing 5/6 nephrectomy, and identified higher level of serum creatinine and blood urea nitrogen (BUN) in CKD rats than in wild-type, depending on time. In addition, the level of α-smooth muscle actin (α-SMA) and collagen I for renal tissue was markedly elevated, with worsening fibrosis due to renal failures. The level of anxiety and depression-like behaviors increased in the 10-week CKD rat models compared with the 4-week rat models. In the recording of local field potentials, the power of delta (1–4 Hz), theta (4–7 Hz), and alpha rhythm (7–12 Hz) was significantly increased in the hippocampus of CKD rats compared with wild-type rats. Together, our findings indicated that anxiogenic behaviors and depression can be induced by CKD, and these abnormal symptoms can be worsened as the onset of CKD was prolonged. In conclusion, our results show that the hippocampus is vulnerable to uremia.

## 1. Introduction

Increased prevalence of chronic kidney disease (CKD) is a substantial public health challenge [1,2,3]. Neurological disorders, such as cerebrovascular disease, cognitive impairment, peripheral neuropathy, and central nervous system dysfunction have been reported during the natural history of CKD [4,5]. In addition, psychological distress and depression are serious challenges in patients with CKD [6,7]. CKD patients who are not on dialysis exhibit three-fold higher rates of depression compared with the general population [8]. Thus, abnormal emotional changes caused by chronic renal failure can have a negative effect on quality of life and clinical outcome [9].

However, previous clinical and animal studies have suggested that CKD itself leads to abnormal brain function and anatomical changes [4,5]. These changes, triggered by CKD, alter behavior and motor functions [10,11,12]. The crosstalk between two organs is mediated by factors including hormones, baroreceptors, and osmoreceptors [4]. Fujisaki and his colleagues suggested that uremia triggers pathologic changes in the hippocampus [13], and impairs synaptic transmission in the rat hippocampus [14]. Further, previous studies showed that long-term CKD affects locomotor activity, anxiety, and depression [10,12,15,16]. However, the mechanisms underlying the altered emotional phenotypes in CKD patients are still unknown [15,16]. Although emotional and psychiatric changes have been observed after progression of CKD, the mechanism and target brain lesions should be defined. In addition, the effect of association with behavior, neuropathological, and abnormal synaptic function in the CKD is unclear.

Electrophysiological studies in brain dysfunctions are interrelated to behavioral tests that act as distributors of brain activity connection with specific behavioral events [17,18]. In particular, previous research using local field potential recordings have already reported that anxiety disorder and depression are linked to hippocampal neural oscillations [19,20,21]. Therefore, we performed local field potential recordings in the hippocampus of a CKD animal model, in order to show: (1) depression-like behavioral phenotypes, and (2) altered hippocampal neural oscillations. Previous studies showed that distinct EEG deviations from normality have cognitive impairments and brain dysfunctions in patients of CKD [22,23]. The exact mechanism of emotional changes is controversial among the different studies, although behavior (lack of social support, burden of illness, and adverse health behaviors) and biological mechanism cause depression in CKD. At least in part, our experimental approaches could explain that anatomical and electrophysiological change in the hippocampus, according to progression of CKD, could be related to altered emotional phenotypes. Thus, this study will be helpful in further research on brain injury caused by CKD. In our present study, the relationship between CKD according to a decline in renal function, and the pathophysiological properties of brain function, such as changes in behavior and electrophysiological alterations, representing emotion-related psychiatric disorders including anxiety and depression, has been directly explored using CKD animal models. Importantly, our results indicate that renal insufficiency is closely related with altered emotional phenotypes, and has a pathogenic effect on brain function following CKD. Finally, our findings will be helpful for extending definitive knowledge of diverse phenotypic sequelae and functional outcomes of CKD.

## 2. Methods

### 2.1. Experimental Animals

All experiments utilized Sprague Dawley rats (8 weeks old) obtained from Experimental Animal Center, Soonchunhyang University (Cheonan, South Korea). All animals were provided with a commercial diet and water ad libitum under controlled temperature, humidity, and lighting conditions (light/dark cycle 12:12, and 22 ± 2 °C, 55 ± 5%). All animal protocols were approved by the Administrative Panel on Laboratory Animal Care of Soonchunhyang University (permit No. SCH18-0056). All possible efforts were made to avoid suffering of the rats and to minimize the number used during the experiments.

### 2.2. CKD Rat Model

Sixty rats were randomly divided into three groups. CKD was induced via 5/6 nephrectomy (5/6 Nx) as described previously with some modifications [24,25]. Briefly, rats were placed in a chamber under general anesthesia using 2.5% isoflurane along with a mixture of 33% oxygen and 67% nitrous oxide. The upper and lower poles of the left kidney were resected using surgical scissors, and sterile gauze was applied to the cut surface for hemostasis for 1–2 min. A week later, a right-sided unilateral nephrectomy was performed. Approximately 5/6 renal mass reduction was achieved. Animals were sacrificed at 4 and 10 weeks post-operatively. Nephrectomized kidneys were used as normal adult kidney controls for immunostaining. As it was an animal experiment, if statistical significance was found, no further experiments were conducted to reduce sacrifice or suffering (Table 1).

### 2.3. Serum Biochemical Assays

Blood samples for standard biochemistry were collected at different time points (4 and 10 weeks after surgery) by cardiac puncture. All serum biochemical assays were performed using commercially available kits. Blood urea nitrogen (BUN) was measured by a specific quantitative colorimetric assay (Quantichrome Urea Assay Kit) developed by BioAssay Systems according to the manufacturer’s protocol. Serum creatinine was measured by another quantitative colorimetric assay (Creatinine Reagent Set), developed by POINTE SCIENTIFIC, following the manufacturer’s instructions.

### 2.4. Behavioral Tests

All animals were tested for behavioral traits of emotional phenotypes at 4 and 10 weeks after the surgery. Behavioral tests were recorded and analyzed using a PC-based video behavioral analysis system using automated tracking software Noldus EthoVision 3.1. In detail, (1) we first examined general locomotor activity and anxiety-like behaviors in the open-field test; (2) 1 day after the open-field test, we measured anxiety-like behaviors in the light-dark test; (3) 1 day after light-dark test, we measured anxiety-like behaviors in the elevated plus-maze test; and (4) lastly, 1 day after elevated plus-maze test, we performed the forced swim test to measure depression-like behaviors.

#### 2.4.1. Anxiety Tests

Three anxiety behavioral assays—open-field, elevated plus-maze, and light-dark transition—were performed between 9:00 A.M. and 5:00 P.M. using adult rats (8- to 16-week-old). Open-field test for anxiety was performed, as described previously [16,26]. Rats were placed in the central square region (60 × 60 cm) of an open-field apparatus (60 × 60 × 40 cm) under diffuse lighting. The extent of their spontaneous movement over the course of 30 min was analyzed using EthoVision 3.1 software (Noldus Information Technology, Wageningen, The Netherlands). The locomotor activity was determined by total distance moved, and anxiety level was measured by the time spent in the central zone. The light-dark transition test was performed as described previously [16,27]. The apparatus consisted of a cage (30 × 30 × 30 cm) divided into two compartments by a black partition containing a small opening that allowed the mouse to move between compartments. One of the boxes was darkened; the other was brightly illuminated. Rats were placed in the illuminated box and allowed to move freely for 5 min. The total time spent in the white compartment served as an indicator of anti-anxiety behavior. In the elevated plus-maze test, rats were placed in an elevated (60 cm above floor level) plus-maze with two opposite open arms (50 × 10 cm each) and two opposite closed arms (50 × 10 cm each) with 50 cm-high walls. The number of entries into and the time spent on individual arms were measured for 5 min. Elevated plus-maze test was performed as described previously [16].

#### 2.4.2. Forced Swim Test

Forced swim test was performed to measure depression-like behaviors as described previously [28,29]. Rats were individually placed in a plastic cylinder (50 cm in height, 30 cm in diameter) filled with water (23 ± 3 °C). Their behavior was observed for 5 min, and the immobility time was calculated. The time spent immobile was considered to reflect depression-like behavior. After the session, the rats were removed from the pool, dried with a towel and returned into their cages.

### 2.5. Local Field Potentials

Local field potentials (LFP) were recorded in the hippocampus of rat brain using previously published protocols with some modifications [30,31]. At the designated times (wild-type, 4 and 10 weeks after surgery), each animal was anesthetized by intraperitoneal injection of urethane 1.5 g/kg and placed in a stereotaxic frame. Holes were drilled through the skull for the introduction of electrodes. In all animals, glass microelectrodes (microfilament capillary 1.2 outer diameter; 5–10 MΩ) filled with artificial cerebrospinal fluid (ACSF, in mM; NaCl 126, KCl 5, CaCl_2_ 2, MgCl_2_ 2, NaH_2_PO_4_ 1.25, NaHCO_3_ 26, D-glucose 10, pH 7.2) were used. The coordinates (in mm) referenced to bregma were as follows (mm to dentate gyrus): 3.8 caudal, 2.5 lateral to bregma, and 2.9 depth. Signals were recorded with a QP511 AC amplifier (0.1–3000 Hz bandpass, GRASS Technologies, West Warwick, RI, USA) and data were digitized (5 kHz) and analyzed using Clampfit 10.2 (Axon Instruments, San Jose, CA, USA). LFP was monitored for 2 h. Analysis of the single-channel electrical traces was carried out using the Clampfit 10.2 (Axon Instruments, San Jose, CA, USA) software. To analyze changes in normalized power of LFP, the amplitude spectrum analysis of normalized power was estimated by event frequency. The root mean square (RMS) values were used to derive estimates of spectral power (mV^2^) in the 1 Hz frequency bins for each electrode site. Spectral power values were averaged across all epochs within a single base-line, and the resulting power was expressed as mV^2^/Hz. For each subject, Fast Fourier Transform (FFT) of the epochs with a resolution of 0.61 Hz was computed for all electrodes and then averaged. Non-overlapping hamming windows controlled spectral leakage. The FFT power values within each frequency between 1 and 50 Hz were averaged to create 50 non-overlapping <1 Hz frequency bins, as the frequency bands of interest were defined as (24, 49, 50) δ (1–4 Hz), θ (4–7 Hz), and α (7–12 Hz). After LFP recording, renal histology and immunohistochemistry were conducted with a few animal specimens.

### 2.6. Renal Histology

Kidney tissues were fixed in 4% paraformaldehyde solution and processed for paraffin embedding. Tissue sections measuring 5 μm in diameter were obtained for Masson’s trichrome staining. The percentage of histological alterations, such as degree of glomerulosclerosis, were evaluated under high magnification (400×) in 5 to 10 consecutive fields, and the mean percentages of histological change were calculated. To conform interstitial fibrosis, immunolabeling for α-SMA and collagen I antibodies was performed. Endogenous peroxidase was quenched with 0.5% H_2_O_2_ for 30 min and nonspecific binding was blocked with 5% normal goat serum in phosphate-buffered saline. The sections were incubated overnight at 4 °C with primary antibodies against α-smooth muscle actin (α-SMA) (Abcam, Cambridge, UK; diluted 1:200) and collagen I (Abcam, Cambridge, UK; diluted 1:400). After incubation with the primary antibody, the tissue sections were treated with a biotin-conjugated anti-mouse IgG secondary antibody (Vector laboratories, Burlingame, CA, USA) and incubated at room temperature for 2 h. Tissue sections were then incubated with ABC complex (Vector) for 2 h at room temperature, and signals were visualized using the 3,3′-diaminobenzidine (DAB) in 0.1 M Tris buffer and mounted on gelatin-coated slides. Immune reactions were observed using DMRB microscope (Leica, Wetzlar, Germany) and images were captured using a model DP72 digital camera and DP2-BSW microscope digital camera software (Olympus, Tokyo, Japan). To establish the specificity of immunostaining, a negative control test was carried out with pre-immune serum instead of the primary antibody. The negative control showed the absence of immunoreactivity in any structures.

### 2.7. Quantification of Data and Statistical Analysis

Optical fractionation was used to estimate the cell numbers. The technique combines optical dissection counting and fractionator sampling, and is a stereological method based on a well-designed, systematic random sampling method. By definition, the approach yields unbiased estimates of population number. Samples of deep tissue were used (optical dissector height, h, was 5 μm). For quantification of immunodensity, DG was delineated with a 2.5 × objective lens. Each image was normalized by adjusting the black and white ranges of the image using Adobe PhotoShop v. 8.0 (San Jose, CA, USA). Thereafter, 10 areas per rat (250 μm^2^ for each area) were selected, and intensity measurements were represented as the mean number of a 256-level gray scale (using NIH Image 1.59 software). Values of background staining were obtained from the corpus callosum. Optical density values were corrected by subtracting the average values of background noise obtained from five image inputs. All data obtained from the quantitative measurements were analyzed using one-way analysis of variance (ANOVA) to determine statistical significance. Bonferroni’s test was used for post hoc comparisons. A *p*-value < 0.05, 0.01, and 0.001 was considered statistically significant [31,32].

## 3. Results

### 3.1. Generation of CKD Rat Models

To investigate the effects of CKD on emotional disorder, such as anxiety and depression, we generated CKD rat models by performing 5/6 nephrectomy. Behavioral tests for measuring anxiety and depression, recording of local filed potentials in the brain, and histological studies in renal and brains were conducted at 4 and 10 weeks following CKD. First, when we identified CKD in these model rats, we found that CKD rat models showed a significant reduction in body weight as compared with wild-type rats (CKD, *p* = 0.005; one-way ANOVA; CKD 4 weeks, reduced 18%, and CKD 10 weeks, reduced 22% approximately compared with wild-type rats; Figure 1A). The level of serum creatinine in CKD rats was higher than in control rats (CKD 4 weeks, *p* < 0.001; CKD 10 weeks, *p* < 0.001; one-way ANOVA; Figure 1B). In addition, the serum BUN level was remarkably increased in CKD rats compared with wild-type rats (CKD 4 weeks, *p* < 0.001; CKD 10 weeks, *p* < 0.001; one-way ANOVA; Figure 1C).

### 3.2. Histological Analysis of Renal Fibrosis in CKD

Next, we performed histological analysis for α-SMA and collagen I in order to identify renal fibrosis in CKD rats using kidney sections (Figure 2). The kidney tissues of CKD rats stained by Masson’s trichrome showed a significantly higher increase in collagen fibers of the interstitial and glomerular tissue with time, compared with wild-type rats (CKD 4 weeks, *p* = 0.03; CKD 10 weeks, *p* < 0.001; one-way ANOVA; Figure 2A,B). Immunohistochemical results showed very weak labeling of α-SMA and collagen I in the renal tissue of wild-type rats, whereas α-SMA levels of CKD rats showed a significant temporal increase in the cytoplasm of specific tubular epithelial cells compared with normal wild-type (CKD 4 weeks, *p* < 0.001; CKD 10 weeks, *p* < 0.001; one-way ANOVA; Figure 2A,C). Further, collagen I in CKD groups was also highly expressed in the interstitium of CKD rats than in wild-type rats following worsening fibrosis due to renal failure (CKD 4 weeks, *p* = 0.018; CKD 10 weeks, *p* < 0.001; one-way ANOVA; Figure 2A,D).

### 3.3. CKD Increases Anxiogenic Behaviors

To determine whether CKD affects anxiety disorders, we performed three anxiety-related behavioral assays in CKD rat models using a previously described procedure (Figure 3A) [16,26,27]. In the open-field assay, CKD rats showed reduced locomotion in the open-field compared with wild-type rats (CKD 4 weeks, *p* = 0.0014; CKD 10 weeks, *p* < 0.001; one-way ANOVA; Figure 3B,C). In addition, the CKD rats spent less time in the central sector of the open field in a time-dependent manner than in wild-type rats (CKD 4 weeks, *p* = 0.016; CKD 10 weeks, *p* = 0.006; one-way ANOVA; Figure 3D), and exhibited a lower amount of defecation in the open field (CKD 4 weeks, *p* = 0.001; CKD 10 weeks, *p* = 0.001; One-way ANOVA; Figure 3E). In the elevated plus-maze, CKD rats made fewer entries into the aversive open arms compared with wild-type rats (CKD 4 weeks, *p* = 0.01; CKD 10 weeks, *p* = 0.02; one-way ANOVA; Figure 3F). The total number of entries (closed plus open arms) decreased in the CKD rats compared with wild-type rats (CKD 4 weeks, *p* < 0.001; CKD 10 weeks, *p* < 0.001; one-way of ANOVA; Figure 3G). Similarly, in the light/dark box test, CKD rats made fewer transitions from the dark to the light compartment than wild-type rats (CKD 4 weeks, *p* = 0.005, CKD 10 weeks, *p* < 0.001; one-way of ANOVA; Figure 3H). Consistent with this finding, the time spent in the light chamber was significantly shorter in case of CKD rats compared with wild-type rats (CKD 4 weeks, *p* = 0.016; CKD 10 weeks, *p* < 0.001; one-way of ANOVA; Figure 3I). Thus, these results demonstrate that CKD is responsible for anxiogenic behaviors and hypolocomotion in rats.

Next, we performed the forced swim test in CKD rats using a previously described procedure in order to investigate whether CKD induced depression-like phenotypes [28,29]. When we assessed depression-like phenotypes, the CKD rats showed severe depression-like behaviors including increased immobile duration (CKD 4 weeks, *p* = 0.007; CKD 10 weeks, *p* < 0.001; one-way of ANOVA; Figure 4A) and decreased latency to immobility in the forced swim test (CKD 4 weeks, *p* = 0.007; CKD 10 weeks, *p* < 0.001; one-way of ANOVA; Figure 4B). Taken together, our findings indicate that CKD affected depression-like behaviors in rats. Furthermore, these altered emotional phenotypes worsened with prolonged onset of CKD.

### 3.4. Representative Profile of LFP in CKD Rat Model

To examine whether brain rhythms, which are related to emotional states including anxiety and depression, were affected by CKD, we recorded LFP in the CA1 region in the hippocampus of CKD rats under urethane anesthesia, using previously established protocols and criteria [30,31]. Urethane (1.5 g/kg) evoked slow rhythms, delta waves, and spindles in wild-type rats (Figure 5A), as previously reported [33]. In the CKD rats, however, epileptic discharges were observed as large amplitude spikes of irregular sharp waves and multiple spikes (Figure 5A). In addition, power spectral densities of LFP at 1–12 Hz were significantly increased in CKD rats compared with wild-type rats (Figure 5B). Briefly, a histogram quantified the absolute power in three different frequency ranges, including delta (CKD 4 weeks, *p* = 0.008; CKD 10 weeks, *p* < 0.001; one-way ANOVA; Figure 5C), theta (CKD 4 weeks, *p* = 0.004; CKD 10 weeks, *p* < 0.001; one-way ANOVA; Figure 5D), and alpha waves (CKD 10 weeks, *p* = 0.014; one-way ANOVA; Figure 5E) in both CKD and wild-type rats, which revealed increased neuronal oscillations at 1–12 Hz in CKD rats. Furthermore, normalized LFP power in CKD rats was much higher than in wild-type rats (CKD 4 weeks, *p* = 0.008; CKD 10 weeks, *p* < 0.001; one-way ANOVA; Figure 5F).

## 4. Discussion

The present study showed behavioral changes representing increased levels of anxiety and depression, and functional changes in neural oscillations in the hippocampus according to a decline in the renal function in an animal model. These findings provide evidence suggesting that uremia induces depression and emotional changes.

In CKD animal models, the weight gain was small compared to the wild-type. Cachexia could develop in CKD patients due to defective central nervous system control of appetite [34]. Further studies will be needed to reveal whether modification in the level of anxiety and depression could affect the weight loss. In humans, estimated glomerular filtration rate (eGFR) is calculated based on creatinine and the other factors according to each equation formulas. These equation formulas, including CKD-EPI, have been validated through several clinical studies [35,36]. However, in animals, there is a lack of a validated formula for obtaining eGFR. It is hard to implement a human eGFR obtaining formula to that of a rat CKD model. In humans, the effect of CKD on behavior change is more affected by the duration of CKD than GFR. As the progress and underlying diseases of CKD are individually distinct in humans, the eGFR reflects the severity or stage of CKD, rather than the duration of CKD. However, our rat CKD model was universally conducted, which means they experienced the same duration of CKD, unlike human studies with various causes and durations of CKD. Therefore, the creatinine in the 4-week and 10-week rat CKD models is thought to reflect not only severity, but also the duration of the disease. Therefore, when applying the results of animal experiments to humans, this discrepancy that, unlike human research, there is a constant prevalence period in the animal study, should be considered.

Multiple factors including uremic toxin, psychosocial, and economic challenges may be associated with the high prevalence of depression in CKD patients [8]. Biochemical studies have been used to elucidate the mechanisms and pathological changes in the brain of CKD animal models [10,15,37]. However, the results of behavioral changes are disputed due to the diversity of CKD animal models, as well as surgical and adenine diet models [15,16,37,38,39]. Thus, different approaches to CKD induction, duration of CKD, and different behavioral tests do not allow a straightforward comparison of results obtained in respective studies. In addition, discrepancies may be attributed to the CKD model regarding different time points of behavioral testing, reflecting early or advanced stages of CKD. Therefore, we tested 4 weeks (early) and 10 weeks (advanced) after CKD surgery. According to previous studies, changes in emotional behavior tests are insufficient and ambiguous [16,38,39]. In the present study, our findings showed a decline in the exploration activity of locomotor personality, and the reduced time spent in central zone, which suggest increased anxiety, as the open-field test reveals general locomotor activation and anxiety. In addition, the results of the light-dark box and elevated plus maze are similar to those of the open-field test. However, the light-dark box demonstrates aversion to light and preference for the dark, while elevated plus maze is used to test dislike for high and open grounds [16]. Further, the forced swim test reveals enhanced depression. Thus, our results are consistent with previous clinical outcomes demonstrating higher levels of anxiety and depression [40]. Previous studies revealed the association of acidosis and brain damage caused by uremic toxicity [41]. In addition, the acid-base imbalance has been reported to have a significant impact on emotion and cognition [42,43]. Therefore, the additional investigations are necessary in order to verify the distinct mechanism on acidosis and related brain damage caused by uremic toxicity.

Clinical studies showed that electrical power in hemodialysis patients was most prominent in delta, theta and alpha frequencies in the temporal and central brain areas (electrode positions T5, T6, C3, and C4) [22]. Explorative statistical comparison of the two data sets with respect to these brain areas revealed that the increase in electrical power in delta, theta, and alpha frequency bands differed from that of healthy individuals. In addition, the group with moderate hippocampal atrophy showed the highest increase in theta power in the frontal regions, and the alpha2 and alpha3 powers in the frontal and temporo-parietal areas [44]. In the present study, our results showed a marked elevation in synchronous neuronal activity of theta waves at CKD 4 and 10 weeks, and increased alpha waves in the CKD 10 weeks. Theta-frequency synchronization has been observed in human subjects with increased anxiety [45] and enhanced theta rhythm was detected during fear responses and anxiety states, supporting the main role of theta rhythm in the modulation of these states [46,47,48]. Furthermore, asymmetric theta rhythm in the hippocampus is a potential biomarker of depression in humans [49]. Prior studies have shown a change in the hippocampal theta and temporal lobe alpha power with depressive symptom patients [19,50]. Therefore, alteration of hippocampal theta rhythm can be a neural oscillatory marker for levels of anxiety and depression in humans and rodents. These findings suggest that the hippocampus is a vulnerable area for uremia, and increased hippocampal neural oscillations at 4–12 Hz are related to abnormal emotional states, such as anxiety and depression in rodents [20,51]. Recently, alpha rhythm was associated with cognitive phenomena, playing an important role in perceptual learning and information processing [51,52]. Thus, the correlation between cognitive ability and CKD models is key to additional experimentation. Additionally, delta rhythm synchronous neuronal activity in the hippocampus offers a unique LFP signal marker in epilepsy [31]. We also observed altered LFP signals in the CA1 region of the hippocampus in an epilepsy pattern as fast and high-amplitude spikes, which may eventually increase the risk of epilepsy in CKD models [53,54]. Further experimental studies associating epilepsy and CKD are needed to confirm this aspect background of CKD-associated sequela in humans.

In presenting this study, our results could explain the altered emotional change in CKD patients. Thus, the present findings suggest that electrophysiological changes in the hippocampus may occur with the progression of CKD, which is related with the emotional phenotype change in patients with CKD. Therefore, this is clinically suggesting that emotional changes could result from anatomical and electrophysiological changes in patients with CKD, which could be a target for treatment. However, there are some limitations in our study. First, behavior tests could be affected by other factors. The forced swimming test is the common behavior test in rodents to evaluate negative mood. However, the forced swimming test could be affected by reduced locomotor activity. Second, the causes of CKD, including diabetes mellitus and hypertension, are diverse in clinical situations. The underlying disease could affect the emotional changes in patients with CKD. Third, age is an important factor in behavior change in general. Therefore, further additional investigations should reveal the effects of these limitations on emotional alteration in CKD.

## 5. Conclusions

Our findings demonstrated that uremia can cause anxiogenic behaviors and depression-like phenotypes, along with alteration of hippocampal neural oscillations at 1–12 Hz in rodents. Based on these results, further studies on immunohistochemical and electrophysiological markers of CKD-related emotional changes should be facilitated.

## Figures and Tables

**Figure 1 brainsci-11-00882-f001:**
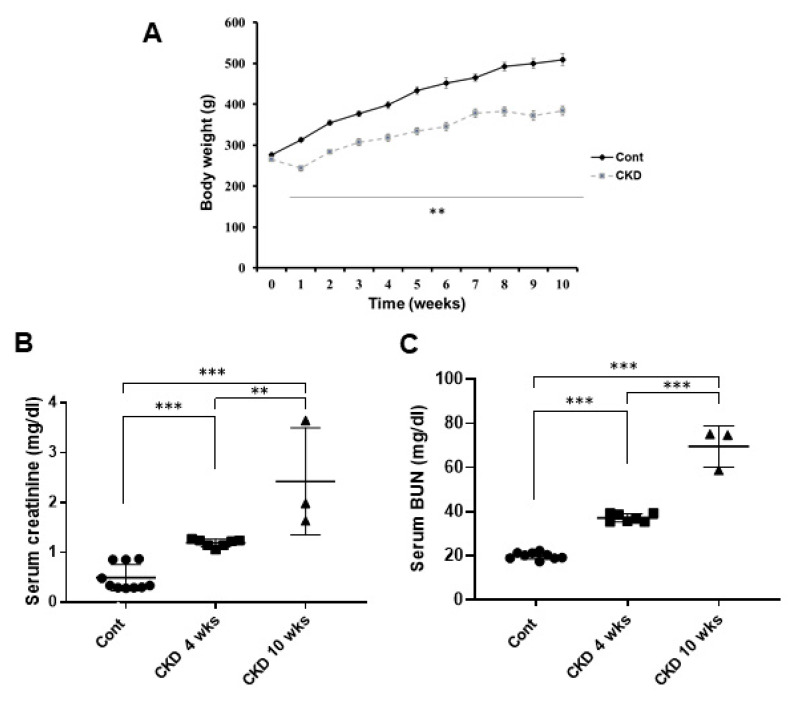
Body weight and blood analysis of CKD and wild-type rat. The body weights of CKD rats were significantly reduced compared with the wild-type group (**A**). Serum creatinine and BUN levels were increased in CKD rat animal model more than in the wild-type (**B**,**C**). Data are presented as means ± standard errors of the mean. ** *p* < 0.01, *** *p* < 0.001, one-way analysis of variance.

**Figure 2 brainsci-11-00882-f002:**
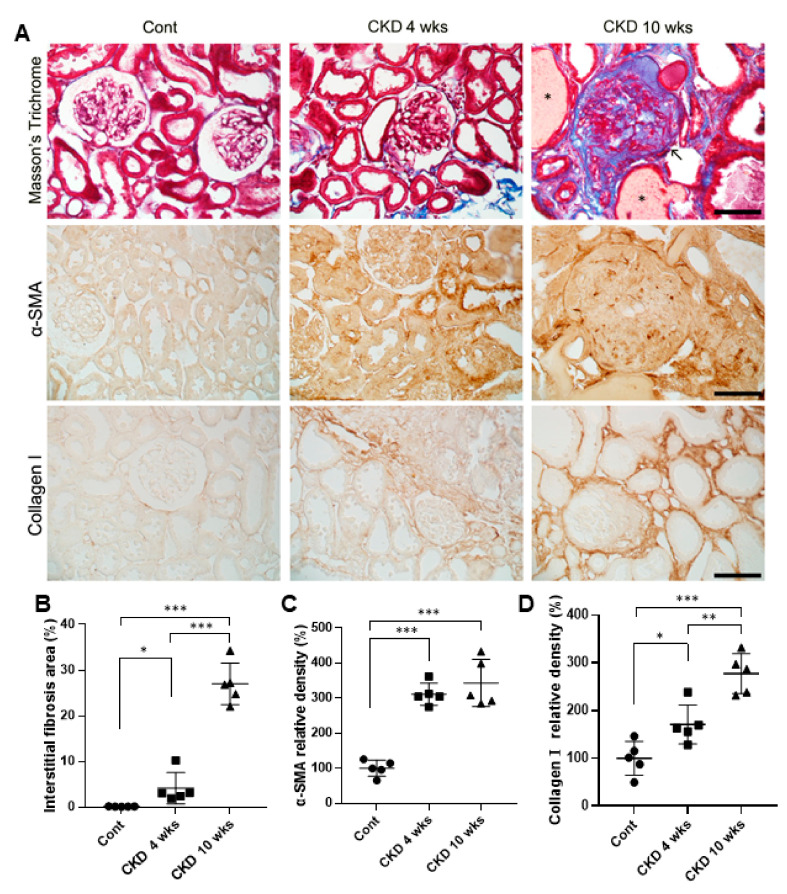
Masson’s trichrome staining, and expression of α-SMA and collagen I in the kidneys of CKD and wild-type rats. Masson’s trichrome staining revealed progressive interstitial fibrosis at CKD 4 and 10 weeks more than in the wild-type (**A**,**B**). 10 weeks-recovered CKD rat models showed glomerulosclerosis collagen deposition in glomeruli (arrows) and injured tubules (*). Morphometric analysis for relative interstitial fibrosis: in CKD rats, increased α-SMA and collagen I immunoreactivities of the interstitium when compared with wild-type rats (**A**,**C**,**D**). In particular, densitometry of collagen I analysis were more enhanced at CKD 10 weeks rats than 4 weeks rats (**D**). Scale bar = 25 μm. Data are presented as means ± standard errors of the mean. * *p* < 0.05, ** *p* < 0.01, and *** *p* < 0.001, one-way analysis of variance.

**Figure 3 brainsci-11-00882-f003:**
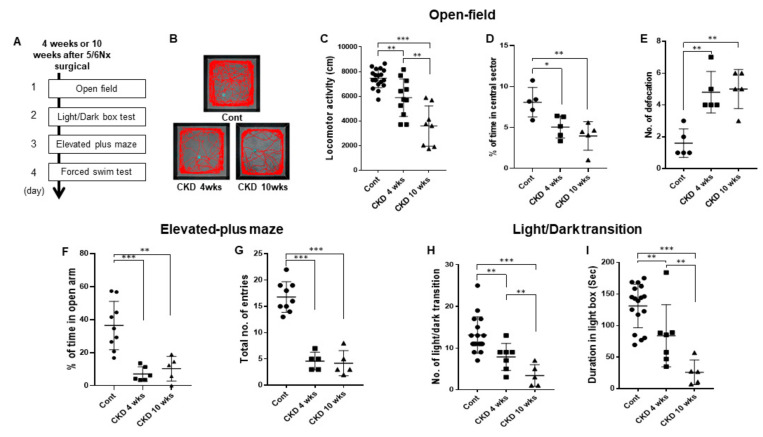
Anxiety-related behaviors in CKD and wild-type rats. Representative diagram shows the overall experimental time schedules (**A**). Representative cumulative traces of navigation pathways of wild-type, CKD 4 week, and CKD 10 week rats during exploratory behavior in the open-field (**B**). Locomotor activity of CKD rat model was decreased as compared with wild-type littermates (**C**). Time spent in the central sector was increased in CKD models and number of defecation was reduced in the open field in CKD groups than wild-type group (**D**,**E**). Percentage of entries into the open arms and total number of entries in the elevated plus-maze were decreased in the CKD models compared with wild-type rats (**F**,**G**). In addition, light/dark transition number and total time in the light were lower in the CKD model than wild-type rats (**H**,**I**). Data are presented as means ± standard errors of the mean. * *p* < 0.05, ** *p* < 0.01, and *** *p* < 0.001, one-way analysis of variance.

**Figure 4 brainsci-11-00882-f004:**
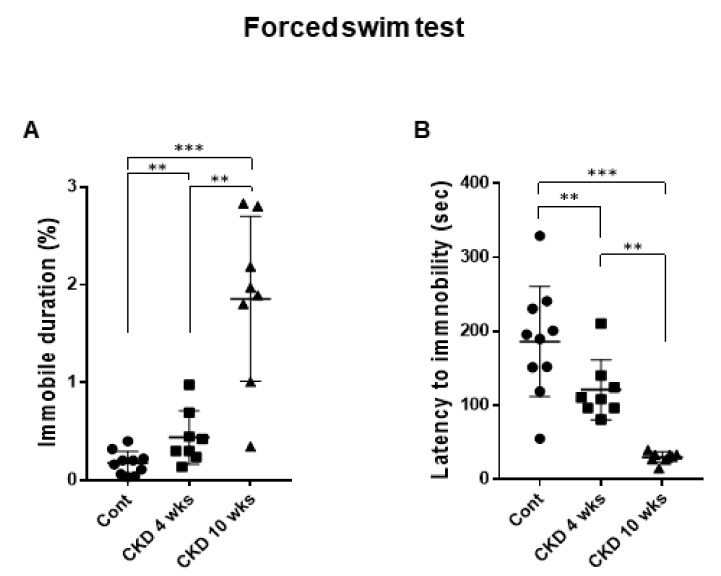
Depression-like behaviors in CKD and wild-type rats. Increased percentages of immobility during forced swim test in CKD rats (**A**). Decreased latency to immobility of forced swim test in CKD rats (**B**). Data are presented as means ± standard errors of the mean. ** *p* < 0.01, *** *p* < 0.001, one-way analysis of variance.

**Figure 5 brainsci-11-00882-f005:**
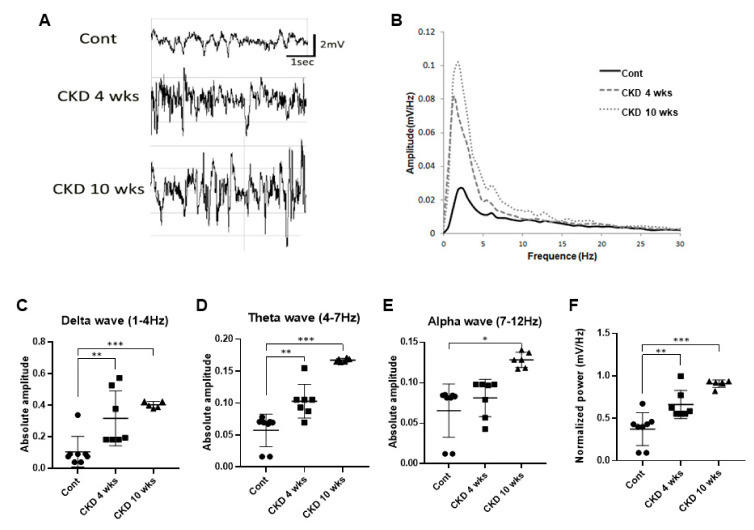
Neural oscillations at 1–12 Hz in the CA1 region of hippocampus of CKD and wild-type rats. Representative LFP signals in the hippocampus of wild-type rats under urethane-induced anesthetic (**A**). The LFP profiles of each CKD rat showed epileptic discharges as large amplitude spikes of irregular sharp waves and multiple spikes (**A**). Power spectral analysis of the CKD rats revealed higher power at a lower frequency more than in the wild-type group (**B**). Hippocampal delta and theta oscillations were markedly elevated in the CKD 4 weeks and 10 weeks-recovered CKD rat model (**C**,**D**), hippocampal alpha rhythm was increased in the 10 week, but not the 4 weeks, recovered CKD rats (**E**). The average normalized power after CKD rats was stronger than in wild-type rats (**F**). Data are presented as means ± standard errors of the mean. * *p* < 0.05, ** *p* < 0.01, and *** *p* < 0.001, one-way analysis of variance (**C**–**F**).

**Table 1 brainsci-11-00882-t001:** Number of tested animals in each behavioral task.

Animal Model	Open-Field	Light-Dark Transition	Elevated Plus-Maze	Forced Swim Test	Blood Analysis	Local Field Potential
Control	23	18	9	9	9	8
CKD 4 weeks	11	9	5	8	7	7
CKD 10 weeks	8	6	5	8	3	5

## Data Availability

All data are available upon request to the corresponding author.

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
