# Peer review of "Altered Emotional Phenotypes in Chronic Kidney Disease Following 5/6 Nephrectomy"

_brainsci, 2021, doi:10.3390/brainsci11070882_

Round 1

Reviewer 1 Report

The present paper reports potentially interesting results. However, the text of the article falls short of giving a satisfactory account, on the one hand, of the aim of the study, and, on the other hand, on the advancement of knowledge in the field provided by the present data.

The previous literature – cited in the paper – shows conflicting results between studies in relation to the ability of CKD to induce anxiety-like and depression-like behaviours. The authors do not provide any explanation on how the results of this study can help to settle the ongoing controversies. Besides, an important problem with the behavioural data is that the increased immobility in the FST is paralleled by reduced locomotor activity. This might question the interpretation of the results of the FST in terms of a depression-like behaviour. Another test, such as the sucrose preference test (as in ref. [16]), might have been useful to deal with this problem (perhaps this problem might be overcome – at least in part – by the discovery of a LFP phenotype which parallels EEG phenotypes observed in human depression, e.g. ref. [19]).

A potential merit of this study might come from the recording of LFP in the hippocampus. However, the interpretation of these results depends on the interpretation of the behavioural results. The most important question to address is: to what extent can we attribute the anxiety-like and depression-like behaviours observed in this study – along with the LFP correlates – to CKD, rather than to the generic exposure to a stressful condition? (E.g.: The result of a study cited in the submitted article [ref. 16] show that a prolonged condition of CKD – with respect to the present study – failed to influence anxiety and depression-like emotional states.)

The present findings do not directly clarify the possible mechanisms by which CKD may induce anxiety-like and depression-like behaviours. At least, a clear and compelling explanation on how they might provide a starting point to address this issue should be proposed.

These problems require a deep rethinking of the whole article text. In particular, the Introduction should provide an explanation on how the experiments performed in this study might help to settle the controversies in regard to the ability of CKD to induce the anxiety-like and depression-like emotional profile. The Discussion should explain to what extent the results of the experiments were successful in this regard. The limitations of the study (e.g. the FST-induced immobility paralleled by reduced locomotor activity) should be discussed.

Author Response

Reviewer1

The present paper reports potentially interesting results. However, the text of the article falls short of giving a satisfactory account, on the one hand, of the aim of the study, and, on the other hand, on the advancement of knowledge in the field provided by the present data.

  1. The previous literature – cited in the paper – shows conflicting results between studies in relation to the ability of CKD to induce anxiety-like and depression-like behaviours. The authors do not provide any explanation on how the results of this study can help to settle the ongoing controversies.

Response:

Of course, with reviewer’s comment, our experimental studies regarding behavior changes in CKD show some controversies. It is reported that functional impairment of short-term memory and a decrease in the number of α4β2 heteromeric neuronal nicotine receptors at 24 weeks after 5/6 nephrectomy (Exp Neurol 2012; 236:28-33.). In addition, there was no difference of behavior characteristics at 3 and 9 months after 5/6 nephrectomy compared with control (Tothva et al., 2015. ref 16). Moreover, behavioural abnormalities and blood-brain barrier disruption increased at 4 weeks after 5/6 nephrectomy (Mazumber et al., 2016. ref 38). On the other hand, rats with adenine-induced chronic renal failure showed decreased motor activity (Ali et al., 2011. ref 15). Thus, different approaches to CKD induction, duration of CKD, and different behavioral tests do not allow a straightforward comparison of results obtained in respective  studies. Therefore, our results in this research can serve as the basis data that the severity of the disease as the duration of the disease passes after CKD is aggravating mental disorders.

So, according to the reviewer’s comments, we newly inserted following sentence in the discussion section as below; 

Thus, different approaches to CKD induction, duration of CKD, and different behavioral tests do not allow a straightforward comparison of results obtained in respective studies (Correction version, Page 16 Line 21).

  1. Besides, an important problem with the behavioural data is that the increased immobility in the FST is paralleled by reduced locomotor activity. This might question the interpretation of the results of the FST in terms of a depression-like behaviour. Another test, such as the sucrose preference test (as in ref. [16]), might have been useful to deal with this problem (perhaps this problem might be overcome – at least in part – by the discovery of a LFP phenotype which parallels EEG phenotypes observed in human depression, e.g. ref. [19]). A potential merit of this study might come from the recording of LFP in the hippocampus. However, the interpretation of these results depends on the interpretation of the behavioural results.

Response:

We appreciate the reviewer’s opinions for raising important comments regarding the involvement of depression. Although the locomotor activity decreased after CKD, the increase in total immobility times despite the decrease in latency of immobility in FST has been regarded as an indication of depression-like behaviour in rodents (Porsolt et al., 2001. ref 21). To be clear, we also performed sucrose preference test, but related research that is being conducted separately. On the other hand, previous electrophysiological studies in rodents using local field potential recording have already reported that anxiety-disorder and depression are related to hippocampal neural oscillations. Especially, asymmetric theta rhythm is a potential brain oscillatory marker of depression in humans (Dharmadhikari et al., 2018. ref 46). In addition, recent studies have shown changes in hippocampal theta and temporal lobe alpha power in the brains of patients with depressive symptoms (Lee et al., 2018. ref 33; Cornwell et al. 2010. ref 19). Therefore, alteration of hippocampal theta rhythm can be a neural oscillatory marker for level of anxiety and depression in humans and rodents.

In our present study, we found depression-like behavioral phenotypes of CKD rat models. Furthermore, in order to examine whether CKD rat model show alteration of hippocampal neural activity, “a neural oscillatory marker of emotional disorder”, we performed local field potential recording in the hippocampus. Notably, we observed that hippocampal oscillations at the frequency of 1-12 Hz including delta, theta, and alpha rhythm, were increased in CKD rat model compared with wild-type rat. These results are consistent with previous reports showing that increased hippocampal neural oscillations at 4-12 Hz are related to abnormal emotional states, such as anxiety and depression in rodents (Gordon et al., 2005. ref 48; McNaughton et al., 2007. ref 20). Therefore, we sincerely hope that the reviewer agrees that this result is helpful for demonstrating the increased depression level of the CKD rat model in our present study.

  1. The most important question to address is: to what extent can we attribute the anxiety-like and depression-like behaviours observed in this study – along with the LFP correlates – to CKD, rather than to the generic exposure to a stressful condition? (E.g.: The result of a study cited in the submitted article [ref. 16] show that a prolonged condition of CKD – with respect to the present study – failed to influence anxiety and depression-like emotional states.)

Response:

We thank the reviewer for this comment. Of course, we agree that stress can affect the experiment. However, it is believed that the statistical differences between 4 wks and 10 wks after 5/6 nephrectomy in our study may show that anxiogenic and depression-like behavior increases related with the duration of CKD. In addition, Tóthová et al [ref 16] showed that compared with our study, there was a very great differences of experimental protocol in the animal species, age and behavior test (one animal repeated the same behavioral analysis experiment; before, 3 months and 9 months after the surgery). Therefore, it is thought that these differences may have influenced the results of emotional states such as anxiety and depression after 5/6 nephrectomy.

  1. The present findings do not directly clarify the possible mechanisms by which CKD may induce anxiety-like and depression-like behaviours. At least, a clear and compelling explanation on how they might provide a starting point to address this issue should be proposed.

Response:

Recent meta-analysis study showed that the pooled prevalence of anxiety disorders (9 studies, n = 1071) among CKD patients across studies was 19% while that of elevated anxiety symptoms (52 studies, n = 10,739) was 43% (Gen Hosp Psychiatry. 2021;69:27-40). In clinical situation, Depression is associated with a substantially increased risk of death in people with CKD (Am J Kidney Dis. 2013 Sep;62(3):493-505). However, as mentioned in previous papers, although CKD is associated with cognitive decline, increased anxiety, or depression, the underlying mechanisms of these changes are still unclear (Tóthová et al, 2015. ref 16). Therefore, the ultimate goal of our study is to elucidate these mechanisms. The first objective of our current study, however, was to investigate whether chronic kidney disease (CKD) animals induces emotional disturbance, such as anxiogenic and depression-like behaviors similarly clinical findings. Previous study revealed the association acidosis and brain damage caused by uremic toxicity (Prog Brain Res. 1993;96:23-48.). In addition, the acid-base imbalance has been reported to have a significant impact emotion and cognition. (Bioimpacts. 2014; 4(2): 53–54, Transl Psychiatry. 2015; 5(5): e572.). Therefore, we in our future investigations are focusing on acidosis and related brain damage caused by uremic toxicity based on other studies.

With respect to reviewer’s comments, we have written the limitations about this topic in discussion part of manuscript as below;

Previous study revealed the association of acidosis and brain damage caused by uremic toxicity. In addition, the acid-base imbalance has been reported to have a significant impact emotion and cognition. Therefore, the additional investigations are necessary in order to verity the distinct mechanism on acidosis and related brain damage caused by uremic toxicity (Correction version, Page 17 Line 5).

  1. These problems require a deep rethinking of the whole article text. In particular, the Introduction should provide an explanation on how the experiments performed in this study might help to settle the controversies in regard to the ability of CKD to induce the anxiety-like and depression-like emotional profile. The Discussion should explain to what extent the results of the experiments were successful in this regard. The limitations of the study (e.g. the FST-induced immobility paralleled by reduced locomotor activity) should be discussed.

Response:

We thank the reviewer for these critically important comments. So we inserted following sentences in the Introduction section for explaining how our study might help to settle these controversies in regard to the ability of CKD to induce emotional profiles as below; .

The exact mechanism of emotional changes is controversial among the different studies although behavior (lack of social support, burden of illness, and adverse health behaviors) and biological mechanism causes depression in CKD. At least in part, our experimental approaches could explain that anatomical and electrophysiological change of hippocampus according to progression of CKD could be related with altered emotional phenotypes. Thus, this study will be helpful in further research on brain injury caused by CKD (Correction version, Page 3 Line 24).

Also, we explained the meaning of our study and limitations in discussion section as below;

In presenting study, our results could explain the altered emotional change in CKD. Thus, present findings suggest that electrophysiological changes of the hippocampus may occur with the progression of CKD, which is related with the emotional phenotype change in patients with CKD. Therefore, this is clinically suggesting that emotional changes could result from anatomical and electrophysiological changes in patients with CKD, which could be a target for treatment. However, there are some limitation in our study. First, behavior tests could be affected by other factors. Forced swimming test is the common behavior test in rodent to evaluated with negative mood. However, forced swimming test could be affected by reduced locomotor activity. Second, the causes of CKD including diabetes mellitus, hypertension are diverse in clinical situation. The underlying disease could affect the emotional changes in patients with CKD. Third, age is an important factor in behavior change in general. Therefore, further additional investigations should reveal the effects of these limitations on emotional alteration in CKD (Correction version, Page 18 Line 4).

Reviewer 2 Report

This is an intriguing study on the effects of uremia on behaviour. The authors used a 5/6th nephrectomy model and studied behaviour using a number of different tests. 

The results of the test are striking. What I am less sure about is whether all of these should be interpreted as depression-type behaviour. E.g. the forced swimming latency time might be explained in different ways (condition? ?Muscle cramps??). 

Author Response

This is an intriguing study on the effects of uremia on behaviour. The authors used a 5/6th nephrectomy model and studied behaviour using a number of different tests.

The results of the test are striking.

  1. What I am less sure about is whether all of these should be interpreted as depression-type behaviour. E.g. the forced swimming latency time might be explained in different ways (condition? ?Muscle cramps??).

Response:

Regarding reviewer’s question, the latency time for the forced swimming test can be affected by other factors. This part may require additional experiments, but nonetheless, the animals showed statistical significance in our experimental results. As well, the increase in total immobility duration despite the decrease in latency of immobility in FST after CKD has been regarded as an indication of depression-like behavior in rodents (Porsolt et al., 2001 ref 28). For more information, please also check our responses to reviewer #1’s similar concerns.